# DNA Aptamers against Vaccinia-Related Kinase (VRK) 1 Block Proliferation in MCF7 Breast Cancer Cells

**DOI:** 10.3390/ph14050473

**Published:** 2021-05-17

**Authors:** Rebeca Carrión-Marchante, Valerio Frezza, Ana Salgado-Figueroa, M. Isabel Pérez-Morgado, M. Elena Martín, Víctor M. González

**Affiliations:** Grupo de Aptámeros, Departamento de Bioquímica-Investigación, IRYCIS-Hospital Universitario Ramón y Cajal, Carretera de Colmenar Viejo Km. 9.100, 28034 Madrid, Spain; rebeca.carrion@hrc.es (R.C.-M.); valerio.frezza@hrc.es (V.F.); ana.salgado@aptusbiotech.com (A.S.-F.); mpmorgado@salud.madrid.org (M.I.P.-M.)

**Keywords:** DNA aptamers, anticancer aptamers, VRK1, breast cancer, cell cycle

## Abstract

Vaccinia-related kinase (VRK) 1 is a serin/threonine kinase that plays an important role in DNA damage response (DDR), phosphorylating some proteins involved in this process such as 53BP1, NBS1 or H2AX, and in the cell cycle progression. In addition, VRK1 is overexpressed in many cancer types and its correlation with poor prognosis has been determined, showing VRK1 as a new therapeutic target in oncology. Using in vitro selection, high-affinity DNA aptamers to VRK1 were selected from a library of ssDNA. Selection was monitored using the enzyme-linked oligonucleotide assay (ELONA), and the selected aptamer population was cloned and sequenced. Three aptamers were selected and characterized. These aptamers recognized the protein kinase VRK1 with an affinity in the nanomolar range and showed a high sensibility. Moreover, the treatment of the MCF7 breast cell line with these aptamers resulted in a decrease in cyclin D1 levels, and an inhibition of cell cycle progression by G1 phase arrest, which induced apoptosis in cells. These results suggest that these aptamers are specific inhibitors of VRK1 that might be developed as potential drugs for the treatment of cancer.

## 1. Introduction

The kinase proteins VRKs (vaccinia-related kinases) are part of serin/threonin-kinases family [1]. The mammalian kinase family of VRK includes three members: VRK1 which is mostly located in the nucleus, VRK2 located in nucleus and endoplasmic reticulum, and VRK3 located in the nucleus [2,3].

VRK1 is located in chromatin covering all DNA except condensed chromosomes [4]. Some proteins involved in DNA damage response (DDR) are phosphorylated by VRK1 such as 53BP1 [5], NBS1 [6] or H2AX [7] which in its phosphorylated state acts as a platform of recruitment for other proteins involved in DDR [8]. VRK1 also forms a complex and phosphorylates p53 mediating protective responses for the cells [9,10]. All these data suggest an important role for VRK1 in the signaling of DNA damage responses [11].

Additionally, VRK1 interacts with numerous proteins involved in cell cycle progression [12] such as CREB [13], mediating cyclin D1 expression and consequently the G1 to S phase transition; the H3 histone [4] resulting in chromatin condensation or BAF [14,15], which is implicated in the nuclear envelope assembly, among others [16,17,18,19,20]. VRK1 is also necessary for cell cycle entry and its loss results in a cell cycle arrest [21], infertility in mice [22] or in a decreased proliferation of a xenograft model of breast cancer [23], indicating an important association of VRK1 with cell proliferation. This association is supported by studies that detect VRK1 expression within the proliferation area of squamous epithelia, as well as Ki-67 expression [24] or high VRK1 levels in tissues with elevated proliferation rates [2].

Moreover, several studies have suggested the involvement of VRK1 overexpression and its correlation with poor prognosis in many cancer types such as head and neck squamous cell carcinoma (HNSCC) [24], lung cancer [25], hepatocellular carcinoma (HCC) [26,27], esophageal cancer [28], glioma [29] or breast cancer [30]. Regarding the latter, VRK1 could promote cancer cell colonization by enhancing the mesenchymal-to-epithelial transition (MET) via downregulating the expression of mesenchymal markers and upregulating that of epithelial markers, which associates VRK1 with breast cancer progression [31]. Indeed, high VRK1 expression shows a significant association with decreased relapse-free survival for patients with ER+ but not for ER- patients [31], indicating that VRK1 overexpression is correlated with poor prognosis in luminal breast cancers [32,33]. The implication of VRK1 in DDR [5,6,7,13,21,24] suggests that it may produce tumor resistance to DNA damaged-based therapies [34] which is consistent with its association to a poor clinical outcome in ER+ breast cancer. These data propose an association of VRK1 with a poor clinical outcome suggesting VRK1 as a potential target in the development of new specific inhibitors in cancer therapy.

Aptamers are single-strand nucleic acids (ssDNA or RNA) that bind with high specificity and affinity to a specific target molecule due to its ability to form three-dimensional structures under certain conditions such as pH, temperature and salinity [35,36,37]. Aptamers selection is performed by the SELEX (Systematic Evolution of Ligands by Exponential Enrichment) [37] procedure which involves carrying out several rounds of selection. In the first round, the initial library with a high diversity of sequences is exposed to the target and the sequences not bound are removed. The bound sequences are eluted and amplified by PCR obtaining an enriched population of double-strand aptamers that will be denatured and then refold before starting the next round of selection. Due to performing successive cycles of selection and amplification, the initial oligonucleotide library is reduced in a few specific sequences towards its target. Selected aptamers are cloned and sequenced in order to study their biological effect. Aptamers bind specifically to their targets obtaining dissociation constants in the nanomolar range. Furthermore, aptamers present many advantages over antibodies since aptamers are smaller displaying better access to their target, are not immunogenic, have increased stability and are rapidly produced in vitro reducing costs [38,39].

We firstly note that the MCF7 (breast adenocarcinoma) cell line has been the object of different in vitro cancer cell studies and apoptosis [40,41]. In our work, we initially performed the isolation and characterization of three aptamers that specifically recognized the protein kinase VRK1 with an affinity in the nanomolar range. Furthermore, we treated the MCF7 cells with these aptamers, which resulted in a decrease in cyclin D1 and retinoblastoma protein levels preventing the cell cycle progression by G1 phase arrest and inducing apoptosis. These preliminary results suggest that our aptamers are specific inhibitors of VRK1 that might be used in cancer treatments.

## 2. Results

### 2.1. Selection of High-Affinity Aptamers against VRK1

Aptamers were selected from libraries of oligonucleotides through iterative cycles of selection (SELEX methodology). We obtained specific aptamers against the VRK1 protein as indicated in the Materials and Methods section. We performed six rounds of selection using a Ni-NTA resin that binds the recombinant VRK1 protein fused to the 6xHIS tail in which the protein was incubated with the initial library in physiological conditions. ELONA assays were performed to check the enrichment of the populations showing an increase in the signal from the population obtained after round 6 (Rd6), relative to population obtained after round 3 (Rd3) and to initial RND35 population (Figure 1A). Since these results, Rd6 was cloned and individual aptamers were isolated and characterized. We obtained 10 aptamers and 9 different sequences (Appendix A). The nine aptamers were checked by ELONA to analyze which one showed the highest affinity to VRK1, showing that sequences corresponding to apVRK8, apVRK28 and apVRK33 reach values at least twice the value of the control (Figure 1B).

After the polymerase chain reaction (PCR), double-strand DNA is obtained, using both chains during the selection process, named the F and R chains. Since the apVRK8, apVRK28 and apVRK33 sequences showed the highest affinity to VRK1, each of their chains were labeled with digoxigenin by PCR separately. As Figure 1C shows, aptamers apVRKF8 (V#F8), apVRKF28 (V#F28), apVRKR28 (V#R28) and apVRKF33 (V#F33) had the highest binding capacity to their target. However, the analysis by ELONA of the four aptamers chemically synthesized (purchased from IBA) indicated that apVRKR28 had lower binding capacity to VRK1 than the others (Appendix A).

### 2.2. Bioinformatic Analysis of the Secondary and Tertiary Structures

Secondary structure prediction was performed on aptamers apVRKF8, apVRKF28, apVRKR28 and apVRKF33 using the mFold software and the QGRS Mapper program to predict G-quadruplexes in nucleotide sequences which strongly stabilizes tertiary structures. Figure 2 shows the most probable secondary structure of these aptamers according to its lower free energy (ΔG). Furthermore, analysis showed that apVRKF8 and apVRKF33 could form two G-quadruplexes structures, while apVRKF28 could form only one. According to both software, apVRKR28 would be the less stable aptamer taking into account its ΔG and the lack of G-quadruplex structures.

Due to all these results (2.1 and 2.2) the aptamers apVRKF8, apVRKF28 and apVRKF33 were used in future studies and apVRKR28 was discarded.

### 2.3. Characterization of the Aptamers Obtained against VRK1

To study the affinity of the three aptamers apVRKF8, apVRKF28 and apVRKF33 for VRK1, we performed ELONA assays in which recombinant VRK1 protein was incubated with increasing concentrations of digoxigenin-labeled aptamers. Data obtained were analyzed by nonlinear regression showing a response to a hyperbola whose equation is y = (x × Bmax)/(x + Kd), where Bmax is the maximal binding and Kd (dissociation constant) is the concentration of ligand required to reach half-maximal binding. Results (Figure 3A) show that apVRKF8, apVRKF28 and apVRKF33 are capable of detecting VRK1 in a concentration-dependent manner with a Kd of 45.82 ± 5.01 nM, 60.06 ± 7.05 nM and 128.8 ± 55.56 nM, respectively.

To study the sensibility of aptamers, we analyzed the minimum amount of protein that is able to detect a fixed concentration of aptamer by Slot blot (Figure 3B). Increasing amounts of VRK1 (0–20 ng) were incubated with two fixed concentrations of aptamers as described in the Materials and Methods Section. Results show that the three aptamers are able to bind to their target detecting at least 5 ng of VRK1 protein. The higher intensity in apVRKF8 aptamer confirms that this aptamer has the highest binding capacity to VRK1 which is in line with the previous results obtained by ELONA.

It is essential to study the high susceptibility of aptamers to enzymes in plasma for their future clinical applications and, therefore, to know if aptamers would need subsequent modifications taking into account their possible therapeutic future. Serum nuclease susceptibility of VRK1 aptamers was performed incubating 300 ng of aptamer in human plasma at 37 °C for 48 h, taking aliquots at times indicated in Figure 3C. Results show that both aptamers apVRKF8 and apVRKF28 are the most stable even after 24 h in the presence of plasma while the apVRKF33 is degraded earlier.

### 2.4. VRK1 Aptamers Block Proliferation in MCF7 Breast Cancer Cells

Once we obtained specific aptamers against VRK1, other objective of this work was to study whether or not these aptamers could inhibit tumor progression in cells. First, we investigated the lipofectamine-mediated delivery of the three aptamers into the breast cell line MCF7. As Figure 4A shows, the three aptamers are able to enter in the cells, being more efficient the entry of apVRKF28, 48 h post-transfection. After 72 h, the intracellular accumulation of the three aptamers is similar. Then, we analyzed the effect of the VRK1 aptamers in cell proliferation and cell cycle.

We analyzed cell viability in MCF7 cells by MTT assay 48 h after transfection with aptamers (Figure 4B). Results showed that the three aptamers are able to decrease cell viability at the two concentrations tested (50 and 100 nM) compared with non-transfected cells, suggesting an antiproliferative effect of these aptamers.

Next, we determined the effect of the aptamers in cell cycle distribution. MCF7 cells were transfected and 72 h after being studied by flow cytometry. As Figure 4C shows, both apVRKF8 and apVRKF28 aptamers significantly increased population in G1 phase (4.6 and 6.5%, respectively) compared with control cells, suggesting that apVRKF8 and apVRKF28 induce an arrest in the G1 phase of the cell cycle. Taking into account that the intracellular accumulation of the aptamer V#F28 at 48 h is the highest of the three aptamers (Figure 4A), we analyzed its effect in the cell cycle at this time showing an increase in 4.7% in G1 phase compared to control cells (Appendix A). In addition, the percentage of cells in subG1 phase increased for the three aptamers being statistically significant for apVRKF8 and apVRKF28 aptamers compared with control no-transfected cells (Figure 4D) indicating that the three aptamers are able to induce apoptosis in MCF7 cells.

To confirm the induction of apoptosis, we analyzed the apoptotic marker cleaved-PARP. Aptamer-transfected MCF7 cells were lysed after 24 h of transfection and analyzed by Western blot as described in the Materials and Methods section. Figure 4D (top) shows PARP was cleaved in these cells proving the apoptotic effect of the three aptamers.

Previous data indicated the implication of VRK1 in cell cycle progression by inducing cyclin D1 expression. In order to inquire about the mechanism of action of our aptamers, we analyzed the effect of them in the cyclin D1 levels. A total of 24 h after transfection, MCF7 cells were lysed and analyzed by Western blot, showing that V#F28 significantly reduces cyclin D1 levels (15%) (Figure 5A). In parallel, we analyzed the levels of phosphorylated and total retinoblastoma (Rb) using an antibody against total Rb protein. The phosphorylated Rb, which migrates slowly in SDS-PAGE, did not change after treatment with the three aptamers (Figure 5B and Appendix A) while the three aptamers reduced at about 20–25% the levels of total Rb protein, being significant for apVRKF8 (Figure 5B). In addition, cyclinD1 and Rb protein levels correlated significantly (Spearman r = 0.721, *p* = 0.003). However, the levels of the proliferation marker PCNA did not change after the treatment with the aptamers (Appendix A). Interestingly, the three aptamers produced a reduction of 20% in VRK1 levels (Figure 5C). Altogether our findings support that VRK1 aptamers inhibit VRK1 activity and levels producing a reduction of cyclin D1 expression and arrest in G1 phase of the cell cycle, preventing cell cycle progression, which leads to a programmed cell death.

## 3. Discussion

VRK1 is a serin/threonine kinase involved in cell cycle progression. Its overexpression has been correlated with a poorer progression in several types of cancer becoming a promising target in the research of new anticancer therapies. In this study, we obtained aptamers that specifically recognize and inhibit VRK1 reducing cyclin D1 levels, which produces a stop in G1-S transition of cell cycle and leads cells to apoptosis.

The SELEX process consisted of six rounds obtaining in the sixth an enriched aptamer population that was cloned and sequenced. By ELONA, apVRKF8, apVRKF28, apVRKR28 and apVRKF33 showed the highest affinity to VRK1 and bioinformatic analysis indicated secondary structure prediction and possible G-quadruplex. Due to the possible lack of G-quadruplex structures in the apVRKR28 aptamer, it was discarded from further studies. The most promising aptamers seemed to be apVRKF8, apVRKF28 and apVRKF33 since they showed the lowest free energies and the possibility of G-quadruplex formation, which confers a significant stability for future applications of aptamers in biological systems. Currently, there are several G-quadruplex aptamers which possess a huge therapeutic potential such as AS1411 that targets nucleolin [42] producing cytotoxic effects on cancer cells [43,44] or T40214 that produces antitumorogenic effects in xenografts models [45].

Later characterization showed that apVRKF8 has the higher affinity to VRK1 both by ELONA and by Slot blot, the three aptamers still being able to recognize their target even at lower concentrations of it.

For future therapeutic-applications it is important that these aptamers are stable for entering into cells and reaching their target without being degraded. In fact, stability assays in plasma showed that apVRKF33 was the most susceptible despite being able to form more G-quadruplex structures than apVRKF28, taking into account the predictive character of bioinformatics analysis. All these results would position apVRKF33 as the aptamer with the lesser affinity for VRK1 and as the less stable.

Aptamers have potential pharmacological advantages such as stability in temperature, pH and ionic concentrations variations [46]. Moreover, aptamers present lack of immunogenicity and a small size which allows them an easier entry into cells [47]. These characteristics provide high potential for aptamers as future diagnostic tools and in cancer treatment [48]. An important characteristic to consider in antitumor-drugs discovery is the inhibition of tumor development processes such as cell proliferation and apoptosis. Accordingly, we demonstrated that apVRKF8, apVRKF28 and apVRKF33 reduced viability in MCF7 cells principally through blocking cell cycle progression. Flow cytometry assays showed that G1 population was increased after apVRKF8 and apVRKF28 transfection and consequently S population was reduced. These data suggest that both aptamers block G1 to S phase transition, which would explain the reduction in cell viability observed by MTT assays. Furthermore, the percentages of sub-G1 phase (apoptotic cells) were significantly increased after MCF7 cells were transfected with apVRKF8 and apVRKF28. At the same time, we found that PARP was cleaved in these cells in response to aptamer transfection, which confirms that apVRKF8 and apVRKF28 induce apoptosis in MCF7 cells.

Several studies suggest that VRK1 plays a key role in cell cycle progression. VRK1 regulates the expression of cyclin D1 [13]; its expression correlates with the proliferation marker Ki-67 in HNSCC [24] and its loss causes a cell cycle arrest [21]. VRK1 might act as an important cell cycle regulator contributing to a poorer tumor prognosis [12]. In HCC [26] and glioma [29], its depletion by siRNAs induces a stop in the G1 phase. These results were obtained with our data (Figure 4C) demonstrating that our aptamers are able to inhibit cell cycle progression in G1-S phase transition. The loss of VRK1 is accompanied by a reduction in cell cycle progression markers such as cyclin D1 an phosphorylated Rb as well as the proliferation marker PCNA indicating an early block in the G1 phase [13,21,24]. The aptamer apVRKF28 significantly reduces cyclin D1 (15%), Rb (26%) and VRK1 (29 %) levels. These results support that at least one of the three selected aptamers against VRK1 is blocking the cell cycle progression in MCF7 cells.

Due to the implication of VRK1 in cell cycle, it can be assumed that working on a synchronized cell culture would improve the effect seen after aptamer transfection. However, we synchronized MCF7 cell cultures and we did not observe differences in cell viability by MTT assays (data not shown) and, consequently, the following studies were performed with a non-synchronized cell culture.

Taking into account that VRK1 phosphorylates some proteins involved in DDR, we analyzed the effect of our aptamers in NBS1 and H2AX (data not shown). A decrease in NBS1 levels and phosphorylated H2AX (p-H2AX) would be expected since our aptamers inhibit VRK1. Although we did not observe changes in NBS1, results showed an increase in p-H2AX of transfected cells compared with control cells. It has been described that p-H2AX is an apoptosis marker [49] so the increase in p-H2AX observed confirms the apoptotic effect of the aptamers in MCF7 cells, which could be mediated by other proteins implicated in H2AX phosphorylation [50].

On the other hand, it could be interesting to confirm the effect of the aptamers in other breast cell lines. In addition, further studies should be conducted on the effect of our aptamers in other VRK1 substrates such as H3 histone [4] or other cell cycle related proteins such as p27 [51], cyclin A and cyclin B1 [24].

In summary, we propose apVRKF8 and apVRKF28 as two ssDNA aptamers that specifically recognize VRK1 protein. Preliminary results indicate that both aptamers significantly reduce cell viability and produce an impairment in the G1/S transition through reducing cyclin D1 levels, which leads to apoptosis. These findings open the possibility to further research in which aptamers can be used in cancer therapy.

## 4. Materials and Methods

### 4.1. Materials

Products were obtained from Sigma (St. Louis, MO, USA) except those indicated in the text. Both non-labeled and labeled primers, RND35 starting population, individual’s DNA aptamers, and their derivatives were purchased from IBA GmbH (Göttingen, Germany).

### 4.2. In Vitro Selection Procedure

Iterative rounds of selection and amplification of ssDNA aptamers were performed in order to obtain aptamers for recombinant VRK1 (Figure 6). In brief, synthetic random library of ssDNA that contains a central randomized region of 35 nucleotides flanked by two conserved 19- and 20-nucleotide regions in each end (5RND35, 5′-ACGCTCGGATGCCACTACAG-35N-CTCATGGACGTGCTGGTGA-3′) was denatured at 95 °C for 10 min and then cooled on ice for 10 min. For the first SELEX round, 25 µg (1 nmol) of RND35 were mixed with 4.7 µg (100 pmol) of VRK1 in 200 µL of selection buffer (10 mM sodium phosphate, 0.15 M NaCl, pH7.5 (PBS), MgCl_2_ 1 mM, BSA 0.2%) and incubated at 37 °C for 1 h. The bound aptamer–VRK1 complexes were purified using Ni-NTA superflow (Qiagen, Hilden, Germany) and the ssDNA bound to the protein were amplified by PCR using the primers named F5 (5′ ACGCTCGGATGCCACTACAG 3′) and R5 (5′ TCACCAGCACGTCCATGAG 3′) under the following conditions: 0.4 µM each primer, 200 µM dNTPs, 2 mM MgCl_2_, and 1 U Taq DNA polymerase (Biotools, Madrid, Spain) in a final volume of 100 µL. The mixture was thermally cycled 20 times through 95 °C for 20 s, 66 °C for 15 s and 72 °C for 30 s. Double-strand PCR product was ethanol-precipitated, and the new ssDNA population was then obtained and structured by heating the dsDNA at 95 °C for 10 min and cooling on ice for 10 min. After rounds 3 and 6, the pool was amplified and labeled using digoxigenin-labeled primers F5 and R5.

### 4.3. Enzyme-Linked Oligonucleotide Assay

ELONA assays were performed to study the enrichment of the populations obtained after each round of selection and the affinity for the target of the individual aptamers as described by Frezza et al. [52]. Aptamers were labeled by PCR using 5′ digoxigenin-labeled F5 or R5 primers or digoxigenin-labeled aptamers purchased by IBA, respectively. Recombinant VRK1 protein was diluted in coating solution (KPL Seracare, Milford, MA, USA) and incubated in a 96-well microtiter plate (NUNC, Roskilde, Denmark) overnight at 4 °C. Afterwards, the wells were blocked with BSA 5% in PBS for 1 h and washed 4 times in selection buffer. Later, digoxigenin-labeled aptamers were diluted in selection buffer at the concentrations indicated in the figure legends, denatured for 10 min at 95 °C and cooled for 10 min on ice to acquire the proper structure. Then, we added 100 µL of the solution to each well and the plate was incubated at 37 °C for 1 h. Later, wells were washed 4 times with selection buffer to remove unbound ssDNA and incubated in the presence of 100 µL of anti-digoxigenin antibody conjugated with horseradish peroxidase (POD) (Roche, Basel, Switzerland) diluted 1/1000. Following 1 h incubation at room temperature on a shacking platform, the plate was washed 4 times and developed using ABTS solution (Boehringer Mannheim, Ingelheim am Rhein, Germany) according to the manufacturer’s instructions. Optical Density (OD) was measured at 405 nm using a TECAN microplate reader.

### 4.4. Aptamer Cloning, Sequencing and Secondary and Tertiary Structure Prediction

Rd6 was amplified by PCR as in 4.2. The dsDNA product with ‘A’-overhangs was cloned into pGEM-T Easy-cloning vector (Promega, Madison, WI, USA) following manufacturer’s instructions. Individual clones were sequenced using T7 (5′-TAATACGACTCACTATAGGG-3′) and Sp6 (5′- ATTTAGGTGACACTATAGAA-3′) primers. Secondary structure predictions were done using the ‘mfold’ software (http://unafold.rna.albany.edu/?q¼mfold/DNA-Folding-Form accessed on 1 May 2016) [53] with modified settings according to the experimental conditions. Additionally, the web-based server QGRS-mapper (http://bioinformatics.ramapo.edu/QGRS/index.php accessed on 1 May 2016) was used to predict the presence of G-quadruplexes in nucleotide sequences. Tertiary structure prediction was done using RNAcomposer software (http://rnacomposer.cs.put.poznan.pl/ accessed on 1 May 2016).

### 4.5. Aptamer Stability Assay

Aptamers (300 ng) were incubated with 10 µL of human plasma at 37 °C. Samples were collected at 0, 6, 16, 24 and 48 h and incubated at 65 °C for 10 min to inhibit nucleases activity and for 10 min on ice. Samples were running on a 3% concentration MS-8 Agarose gel (Conda, Madrid, Spain) in 1 × TAE buffer and visualized wigh GelRed (Biotium, Hayward, CA, USA).

### 4.6. Slotblot

Recombinant VRK1 protein between 0 and 20 ng were transferred onto nitrocellulose membranes under vacuum. Filters were washed three times in PBS-T (PBS, 0.1% Tween-20) for 10 min and then blocked with 5% dry milk in PBS-T for 1 h at room temperature. Afterward, membranes were incubated with structured digoxigenin-labeled aptamers at 20 or 40 nM in selection buffer for 1 h at 37 °C temperature with gentle rocking. After, membranes were washed with selection buffer three times and probed with anti-digoxigenin-POD antibody diluted 1/5000, for 1 h at room temperature, followed of three washes with selection buffer. Finally, the membranes were developed with enhanced chemiluminescence’s kits (GE Healthcare, Madrid, Spain) and exposed to hyper film.

### 4.7. Aptamer Quantification

For the quantification of the intracellular aptamer, cells were pelleted 48 and 72 h post-transfection, washed twice with PBS, lysed in 30 µL of H_2_O by homogenization, vortexed and boiled at 90 °C for 10 min. Afterwards, the lysates were centrifuged at 12,000× *g* for 10 min and the aptamers in the supernatants quantified by quantitative PCR. Quantification was performed with the QUANTIMIX EASY kit (Biotools, Madrid, Spain) in an iQ5 equipment (Bio-Rad, Barcelona, Spain). The reaction mixture consisted of a 1× master mix, 0.2 µM primers (F5 and R5) and 1 µL of template in a 20 µL/tube final volume. The aptamers were quantified using a standard curve for each aptamer (100 fmol–100 amol).

### 4.8. Cell Culture and Transfection

Breast adenocarcinoma (MCF7) cells were grown in DMEM/Ham F-12 medium (Biowest SAS, France) (mixed 1:0.8) supplemented with 10% fetal bovine serum (Gibco, grand island, NY, USA), 1% pyruvate, 1% glutamine, 100 U/mL penicillin, 100 µg/mL streptomycin and 25 µg/mL amphotericin in a humidified 5% CO_2_/95% air incubator at 37 °C. Cells were passaged using trypsin 1× (Gibco) in culture medium. For aptamers transfection, cells were plated at different concentrations according to the assay and after 24 h, aptamers structured as indicated in Section 4.2, were transfected using Lipofectamine^TM^ 2000 (Invitrogen, Carlsbad, CA, USA) according to manufacturer’s recommendations. Cells were incubated at 37 °C in a humidified 5% CO_2_/95% air incubator until the assay was performed (Figure 7).

### 4.9. Protein Extraction, Dodecyl Sulphate-Polyacrilamide Gel Electrophoresis and Immunoblotting

To obtain cell lysates, cells were trypsinized as indicated above and washed once with cold buffer A (20 mM Tris–HCl pH 7.6, 1 mM dithiothreitol (DTT), 1 mM ethylenediaminetetraacetic acid, 1 mM phenylmethylsulfonyl fluoride, 1 mM benzamidine, 10 mM sodium molybdate, 10 mM sodium β-glycerophosphate, 1 mM sodium orthovanadate, 120 mM potassium chloride (KCl), 10 µg/mL antipain, 1 µg/mL pepstatin A, and leupeptin) centrifuging at 400× *g* for 5 min. Next, cells were lysed in the same buffer A containing 1% Triton X-100 (volume ratio 1:2) and centrifuged at 12,000× *g* for 10 min. Protein concentration was determined by BCA kit (ThermoFisher Scientific, Waltham, MA, USA) using the supernatant which was aliquoted and stored at −80 °C until use.

Proteins were separated in a 12% sodium dodecyl sulphate-polyacrylamide gel (SDS-PAGE) by electrophoresis and transferred onto polyvinylidene difluoride membranes. Following, membranes were incubated with 5% non-fat milk in PBS for 1 h at room temperature, then with primary β-actin, VRK1, Retinoblastoma (Sigma, St. Louis, MO, USA), cleaved-PARP (Cell Signaling Technology, Danvers, MA, USA), cyclin D1 (Abcam, Cambridge, UK) or PCNA (Santa Cruz Biotechnology, Dallas, TX, USA) antibodies overnight at 4 °C and secondary antibodies (GE HealthCare) for 1 h at room temperature. After each incubation step membranes were washed 3 times with 0.05% Tween 20 in PBS. Primary antibodies were diluted in 0.1% Tween 20 in PBS and secondary antibodies were diluted in 1% non-fat milk in PBS or in 1% bovine serum albumin (BSA) in PBS. After the final washing step, bound proteins were visualized using Clarity^TM^ Western ECL (Bio-Rad, Barcelona, Spain) and ChemiDoc^TM^ (Bio-Rad). PageRuler Plus Prestained Protein Ladder (ThermoFisher Scientific) was the molecular weight marker used in all the experiments. Quantification of the bands was performed by Image-Lab (Bio-Rad) software and results were obtained in optical density/mm^2^.

### 4.10. MTT Assay

Cells were plated at 10^4^ cells/well in a p96 plate and, after 24 h, aptamers were transfected at increasing concentrations indicated in the figure legends. After 48 h, medium was removed and 100 µL/well MTT 1 mg/mL in culture medium were added and plates were incubated for 3 h at 37 °C. Next, 100 µL lysis buffer (10% sodium dodecyl sulphate and 10 mM HCl) were added to each well and after 24 h of incubation the optical density was read using a wavelength of 540 nm on a TECAN microplate reader Infinite F200. Percent inhibition was calculated relative to the cells transfected with no aptamer (control).

### 4.11. Cell Cycle Assay

To analyze the progression of the cell cycle, cells were plated at 5 × 10^5^ cells/well in a p6 plate and after 24 h aptamers were transfected at 50 nM. After 72 h cells were trypsinized and supernatants were collected too and altogether were washed twice in PBS centrifuging at 400× *g* for 5 min. Cells were fixed in 70% ethanol at −20 °C for 30 min, washed twice in PBS as previously indicated, resuspended in 1 mL PBS containing 50 µg/mL propidium iodide and 100 µg/mL RNase, incubated in the dark for 1 h at 37 °C and analyzed using the FACS Calibur^TM^ cytometer (BD Biosciences, San Jose, CA, USA). Acquired data were analyzed using Flowing software 2.5.1 (Turku Biosciences, Turku, Finland).

### 4.12. Statistical Analysis

Data are presented as an average value ± SEM from three to five independent measurements in separate experiments and analyzed using GraphPad Prism 8 (San Diego, CA, USA). The statistical significance was performed by one sample t-test against a control value. Significance was assumed at *p* < 0.05.

## Figures and Tables

**Figure 1 pharmaceuticals-14-00473-f001:**
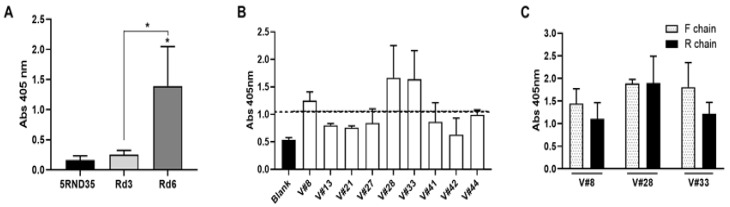
Selection and Identification of aptamers recognizing VRK1 protein. (**A**) VRK1 binding of round 3 (Rd3), round 6 (Rd6) population and RND35 initial pool by ELONA as described in the Materials and Methods Section. Recombinant protein VRK1 (200 ng/well; 4.3 pmol/well) was incubated with digoxigenin-labeled Rd3 or Rd6 aptamer population or digoxigenin-labeled RND35 at 2 ng/µL (80 nM). The graph represents the mean ± SEM of 3 independent experiments (* *p* < 0.05). (**B**) ELONA assay with F and R-digoxigenin-labeled individual aptamers was performed as in (A). The dotted line indicates the double of blank. (**C**) The chain F or R of the selected aptamers apVRK8 (V#8), apVRK28 (V#28) and apVRK33 (V#33) were labeled with digoxigenin and ELONA assay performed as above using the aptamers at a concentration of 1 ng/µL (40 nM). The graphs represent the mean ± SEM of 2 independent experiments.

**Figure 2 pharmaceuticals-14-00473-f002:**
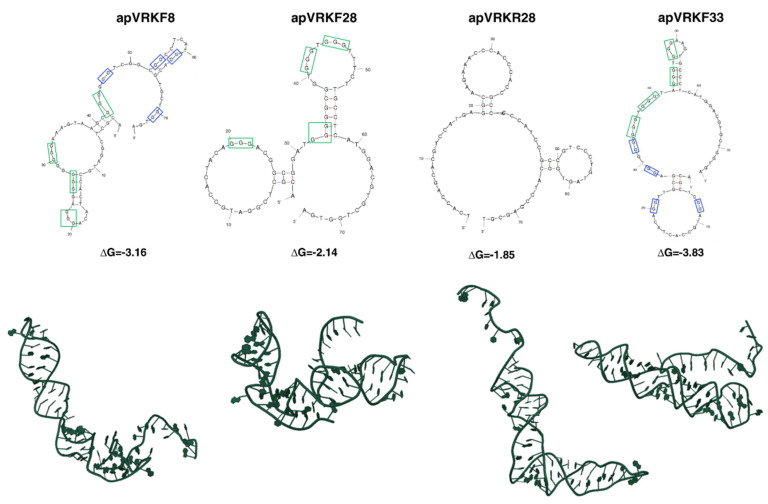
Secondary and tertiary structures of VRK1 aptamers. The structures were obtained using mFold, QGRS mapper and RNAcomposer software. The secondary structures with the lowest free energy and the proposed tertiary structures (centroid fold) are shown. Blue and green squares indicate G-doublets or G-triplets with highest probability (G-Score) of participating to the G-quadruplex formation, based on the algorithm used by QGRS-mapper software.

**Figure 3 pharmaceuticals-14-00473-f003:**
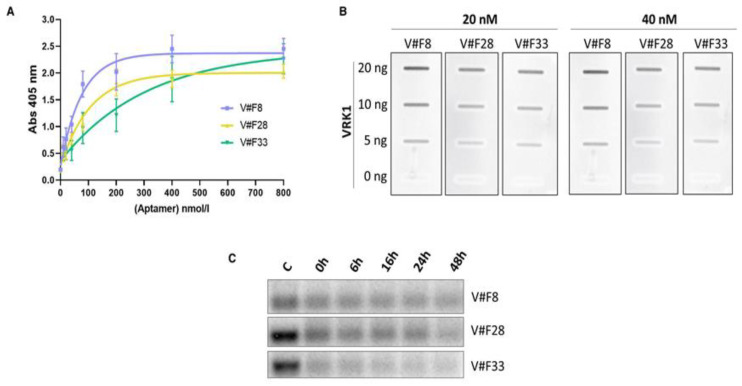
Characterization of VRK1 aptamers. (**A**) ELONA assays to determine the binding affinity of the aptamers to VRK1. The recombinant VRK1 protein was plated at 20 ng/well and then incubated with the digoxigenin-labeled aptamers at increasing concentrations (0–800 nM). All the experiments were performed in triplicate and an average ± SEM of 3–5 different experiments is shown. (**B**) Slot blot in which increasing amounts of VRK1 protein (0–20 ng) were fixed under a vacuum to a nitrocellulose membrane and incubated with digoxigenin-labeled aptamers at 20 and 40 nM. Afterwards, the membrane was probed with anti-digoxigenin-POD, developed with an enhanced chemiluminescence kit and exposed to hyperfilm. The membrane shown is representative of three different experiments. (**C**) Aptamer susceptibility to enzymes in plasma. Representative agarose gels are shown.

**Figure 4 pharmaceuticals-14-00473-f004:**
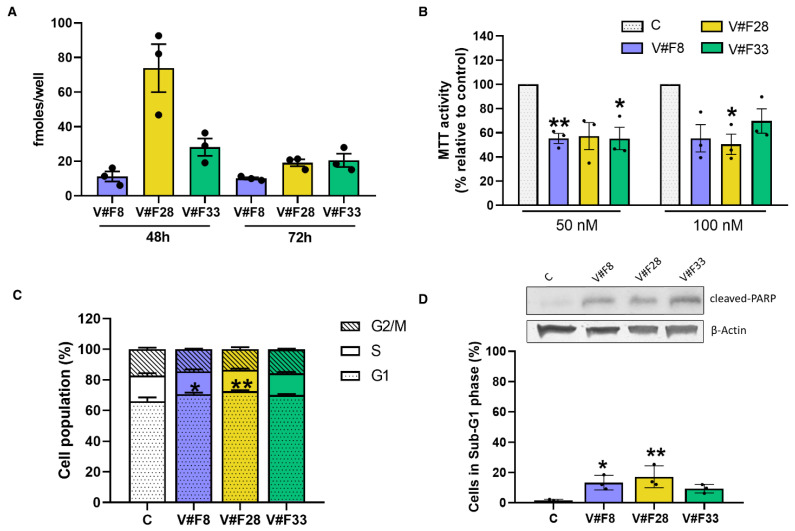
VRK1 aptamers decrease cell viability, arrest cells in G1 phase of cell cycle and increase apoptosis. (**A**) MCF7 cells were plated at 10^4^ cells/well in p96 plates. After 24 h, aptamers were transfected at 50 nM for 48 and 72 h and intracellular accumulation of the aptamers was measured as described in the Materials and Methods Section. (B) MCF7 cells were plated at 10^4^ cells/well in p96 plates. After 24 h, aptamers were transfected at 50 and 100 nM for 48 h, then MTT assays were performed. (**C**,**D**) MCF7 cells were transfected with 50 nM aptamers. After 72 h cells were stained with PI and subjected to flow cytometry analysis. The percentage of cells gated in each phase is indicated. (**D**) Top, Lysates were subjected to SDS-PAGE 12% and Western blot analysis was performed as described in the Materials and Methods Section. Actin was used as a control for the homogeneity of loading. A representative blot is shown. Cells transfected with no aptamer were used as the control. Bars represent the mean ± SEM of 3 independent experiments. Statistical differences relative to the control; * *p* < 0.05; ** *p* < 0.01.

**Figure 5 pharmaceuticals-14-00473-f005:**
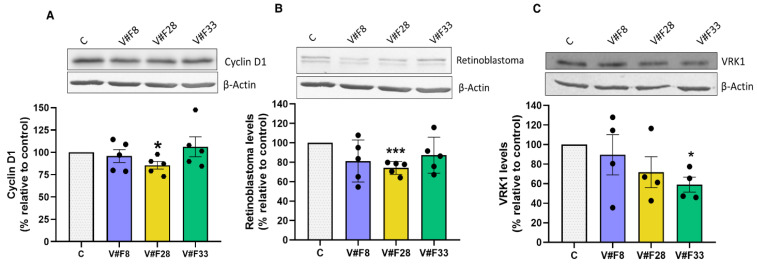
VRK1 aptamers decrease Cyclin D1, Rb and VRK1 protein levels. MCF7 cells were plated at 5 × 10^5^ cells/well in p6 plates. After 24 h, aptamers were transfected at 50 nM for 24 h and cell lysates (20 μg), obtained as described in the Materials and Methods Section, were subjected to SDS-PAGE 12% and Western blot analysis using anti-cyclin D1 (A), anti-Rb (B) or anti-VRK1 (C) antibodies. Actin was used as a control for the homogeneity of loading. The quantitation of the bands was normalized with respect to actin and expressed as the percentage relative to the value in control cells. The values represent the mean ± S.E.M. of 4–5 different experiments. Representative blots are shown on the top of the figure. Statistical differences relative to 100; * *p* < 0.05; *** *p* < 0.001.

**Figure 6 pharmaceuticals-14-00473-f006:**
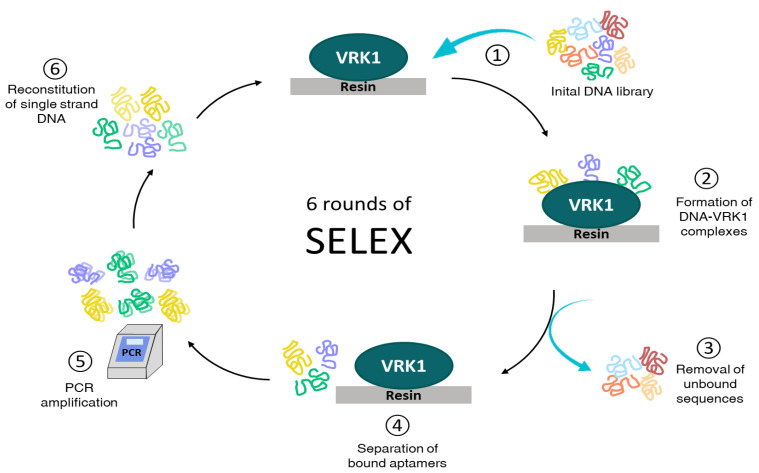
SELEX process used for VRK1 aptamers.

**Figure 7 pharmaceuticals-14-00473-f007:**
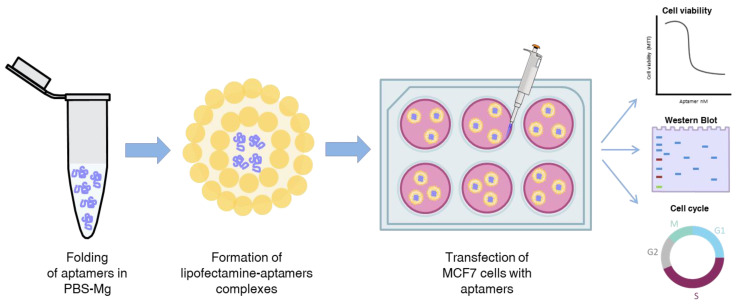
Workflow of aptamers preparation and treatment of MCF7 cells.

## Data Availability

Data is contained within the article or Appendix A.

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
