# Peer review of "DNA Aptamers against Vaccinia-Related Kinase (VRK) 1 Block Proliferation in MCF7 Breast Cancer Cells"

_pharmaceuticals, 2021, doi:10.3390/ph14050473_

Round 1

Reviewer 1 Report

The work sounds very intresting and significant. I highly recommend this article for the publication after some changes.

Line 65: correct English: “binds” to “bind”.

Line 268-269: Could you explain the concentration difference between the library concentration and target? Why the library concentration exceed the target concentration by 10 times.

Line 114: Please check the color correspondence to doublets and triplets (apVRKF28 doesn’t match the description as I understood)

Figure 4B: There is no data distribution for the controls (all three points are at the same level)?

Figure 5: There is no data distribution for the controls (all points are at the same level)? Could it be that the levels of expression of cyclin D1, total Rb protein, VRK1 are same even for the different control experiments. Is there any dispersion? It is important to compare the levels for controls and noncontrols and overlapping of standard deviations.

Line 299: Please correct primers.

Fugure S1: Could you add the blanks for these two experiments?

Author Response

Reviewers #1

The work sounds very interesting and significant. I highly recommend this article for the publication after some changes.

Line 65: correct English: “binds” to “bind”.

It has been corrected as suggested by the referee.

Line 268-269: Could you explain the concentration difference between the library concentration and target? Why the library concentration exceeds the target concentration by 10 times.

The objective of the selection is to recover those aptamers that interact with the target with the highest affinity. By putting 10 times less target, the aptamers will compete with each other to bind to the target, and only those that show greater affinity will do so. If the aptamer:target ratio was 1: 1, we would be allowing, eventually, the interaction of all aptamers.

Line 114: Please check the color correspondence to doublets and triplets (apVRKF28 doesn’t match the description as I understood)

We have changed the color of the triplets in apVRKF28 as indicated by the referee.

Figure 4B: There is no data distribution for the controls (all three points are at the same level)?

Figure 5: There is no data distribution for the controls (all points are at the same level)? Could it be that the levels of expression of cyclin D1, total Rb protein, VRK1 are same even for the different control experiments? Is there any dispersion? It is important to compare the levels for controls and noncontrols and overlapping of standard deviations.

Relative to Figure 4B and Figure 5, in each western blot, we loaded the samples corresponding to an experiment including control and aptamers-treated cell lysates. We have measured the intensity of the bands using image lab software (BioRad) and normalized relative to control levels, which are considered 100%, for each experiment. For this reason, all the controls have a value of 100. Although the bar could be deleted, they have been included as a reference for the value 100. In this new version, the points in the control bars have been deleted. On the other hand, the levels of cyclin D1, Rb or VRK1 in control cells run in the same gel did not vary between them.

Line 299: Please correct primers.

Primers have been corrected and highlighted.

Figure S1: Could you add the blanks for these two experiments?

We have included the blanks in Figure S1 as suggested by the referee.

Reviewer 2 Report

Is interesting paper about of the DNA Aptamers as specific inhibitor of VRK1 in breast cancer.    First, the use of English was clear and understandable. Referring to the methods they were clear, logical, and on line with the project questions. However some points could be discussed.   1) In the cell cycle analysis I recommend using stacked bar graphs where we can notice the proportions of cells in the different stages of cell cycle per group of treatment.    2) In the introduction the authors mentioned that one of the main mechanisms in which VRK1 could promote cancer cell colonization is by enhancing the mesenchymal to epithelial transition (MET), via downregulation of the expression of mesenchymal markers and upregulating the epithelial ones. Also, they mentioned its role over DDR. But you don't explore these mechanisms in the experiments nor the discussion. I consider it is important to measure or discuss the effect of the aptamers over these mechanisms.    3) If the efficiency of transfection was better at 48 hours.  Why did study the effect over the cell cycle after 72 hours, instead of 48 hours like in your cell viability assay? I think it would be important to follow the effect of the aptamers over the cell cycle at different times of treatment to be consequent with your other experiments.    In my opinion the paper could be published after minor changes  

Author Response

Reviewers #2

Is interesting paper about of the DNA Aptamers as specific inhibitor of VRK1 in breast cancer.    First, the use of English was clear and understandable. Referring to the methods they were clear, logical, and on line with the project questions. However, some points could be discussed.  

1) In the cell cycle analysis, I recommend using stacked bar graphs where we can notice the proportions of cells in the different stages of cell cycle per group of treatment.  

We have changed the Figure 4C and used stacked bar graph as indicated by the referee.

 2) In the introduction the authors mentioned that one of the main mechanisms in which VRK1 could promote cancer cell colonization is by enhancing the mesenchymal to epithelial transition (MET), via downregulation of the expression of mesenchymal markers and upregulating the epithelial ones. Also, they mentioned its role over DDR. But you don't explore these mechanisms in the experiments nor the discussion. I consider it is important to measure or discuss the effect of the aptamers over these mechanisms. 

We have worked with MCF7 cells which have an epithelial phenotype and a low metastatic capacity. We consider that it would be interesting try to measure the effect of aptamers in MET in another cell type with a higher metastatic capacity. Relative to DDR, a new paragraph has been included in Discussion section (lines 255-260).

 3) If the efficiency of transfection was better at 48 hours.  Why did study the effect over the cell cycle after 72 hours, instead of 48 hours like in your cell viability assay? I think it would be important to follow the effect of the aptamers over the cell cycle at different times of treatment to be consequent with your other experiments.  In my opinion the paper could be published after minor changes

We agree with the referee. Since the intracellular accumulation of the aptamer V#F28 at 48 h is the highest of the three aptamers, we have also analyzed its effect in cell cycle at this time. Results have been included in the text (line 164) and in a new supplementary figure S2.

Reviewer 3 Report

In this paper, DNA aptamers to VRK 1 pivotal in DNA DDR in MCF-7 cells, were studied.

In MCF-7 line, aptamers caused down-regulation of cyclin D1 and inhibition of cell cycle process.

As the ultimate result, such aptamers are suggested to be specific inhibitors associated with WRK1, thus being potential drug compounds to treat cancer progression.

REMARKS

INTRODUCTION (note)

Some of the impact articles to cite are missing. Please, follow the following upgrade.

INTRODUCTION  (upgrade)

We firstly note that MCF7 (breast adenocarcinoma) cell line has been the object for different in-vitro cancer cell studies and apoptosis [https://doi.org/10.1515/biol-2019-0070, https://doi.org/10.1016/j.advms.2020.03.002]. In our work, we initially performed the isolation and characterization of three aptamers that specifically recognized the protein kinase VRK1 with an affinity in the nanomolar range. Further we treated the MCF7 cells with these aptamers resulted in a decrease in cyclin D1 and retinoblastoma protein levels preventing the cell cycle progression by G1 phase arrest and inducing apoptosis.

RESULTS, MATERIALS AND METHODS (notes)

I found these sections parts of manuscript exhaustive, enough in length and easy in presentation.

One thing I miss for sure is to draw and display as a figure, the operational workflow as for the preparation of aptamers and the treatment of MCF7 cells with them. The figure of workflow schematic drawing would serve for the quick glance for the reader to immediate comprehending the aim of the authors´ work.

Author Response

Reviewers #3

In this paper, DNA aptamers to VRK 1 pivotal in DNA DDR in MCF-7 cells, were studied.

In MCF-7 line, aptamers caused down-regulation of cyclin D1 and inhibition of cell cycle process.

As the ultimate result, such aptamers are suggested to be specific inhibitors associated with WRK1, thus being potential drug compounds to treat cancer progression.

REMARKS

INTRODUCTION (note)

Some of the impact articles to cite are missing. Please, follow the following upgrade.

INTRODUCTION (upgrade)

We firstly note that MCF7 (breast adenocarcinoma) cell line has been the object for different in-vitro cancer cell studies and apoptosis [https://doi.org/10.1515/biol-2019-0070, https://doi.org/10.1016/j.advms.2020.03.002]. In our work, we initially performed the isolation and characterization of three aptamers that specifically recognized the protein kinase VRK1 with an affinity in the nanomolar range. Further we treated the MCF7 cells with these aptamers resulted in a decrease in cyclin D1 and retinoblastoma protein levels preventing the cell cycle progression by G1 phase arrest and inducing apoptosis.

Thank you very much for your suggestions, which we have incorporated into the text

RESULTS, MATERIALS AND METHODS (notes)

I found these sections parts of manuscript exhaustive, enough in length and easy in presentation.

One thing I miss for sure is to draw and display as a figure, the operational workflow as for the preparation of aptamers and the treatment of MCF7 cells with them. The figure of workflow schematic drawing would serve for the quick glance for the reader to immediate comprehending the aim of the authors´ work.

We have prepared two new figures: Figure 6 in which we show the SELEX method used in this work, and Figure 7 that shows the preparation of aptamers and the treatment of MCF7 cells with them.

Round 2

Reviewer 3 Report

Authods have accomplished all given remarks. I consider manuscript to be published at current form